# A Dynamic Multi-Scale Convolution Model for Face Recognition Using Event-Related Potentials

**DOI:** 10.3390/s24134368

**Published:** 2024-07-05

**Authors:** Shengkai Li, Tonglin Zhang, Fangmei Yang, Xian Li, Ziyang Wang, Dongjie Zhao

**Affiliations:** 1School of Automation, Qingdao University, Qingdao 266071, China; lishengkai1110@gmail.com (S.L.); lixian@qdu.edu.cn (X.L.); 2State Key Laboratory of Multimodal Artifcial Intelligence Systems, The Institute of Automation, Chinese Academy of Sciences, Beijing 100190, China; zhangtonglin2021@ia.ac.cn (T.Z.); fangmei.yang@ia.ac.cn (F.Y.); 3School of Artificial Intelligence, University of Chinese Academy of Sciences, Beijing 100190, China; 4Shandong Key Laboratory of Industrial Control Technology, Qingdao University, Qingdao 266071, China

**Keywords:** familiar and unfamiliar face recognition, mask, multi-scale

## Abstract

With the development of data mining technology, the analysis of event-related potential (ERP) data has evolved from statistical analysis of time-domain features to data-driven techniques based on supervised and unsupervised learning. However, there are still many challenges in understanding the relationship between ERP components and the representation of familiar and unfamiliar faces. To address this, this paper proposes a model based on Dynamic Multi-Scale Convolution for group recognition of familiar and unfamiliar faces. This approach uses generated weight masks for cross-subject familiar/unfamiliar face recognition using a multi-scale model. The model employs a variable-length filter generator to dynamically determine the optimal filter length for time-series samples, thereby capturing features at different time scales. Comparative experiments are conducted to evaluate the model’s performance against SOTA models. The results demonstrate that our model achieves impressive outcomes, with a balanced accuracy rate of 93.20% and an F1 score of 88.54%, outperforming the methods used for comparison. The ERP data extracted from different time regions in the model can also provide data-driven technical support for research based on the representation of different ERP components.

## 1. Introduction

Self-face recognition reflects the process by which individuals identify their own faces through differentiation from others [1]. Face recognition is widely used in psychological experiments not only to study face perception but also in areas such as socio-emotional decision making, interpersonal relationships, intra-group and inter-group biases, and stereotypes. In recent years, researchers have focused on using event-related potentials (ERPs) to detect self-face recognition [2,3,4]. ERP technology, which, by measuring the electrophysiological response of the brain to specific stimuli or tasks, provides researchers with a refined tool to observe subtle changes in brain activity during the face recognition process. In the study of face recognition, specific ERP components such as N170 have been found to be closely related to the structural encoding of facial features [5]. N170 is a negative wave that appears around 170 ms after the presentation of a face stimulus, and it is primarily observed in the temporal region and considered an early marker of face perception [6]. Additionally, other ERP components, such as P300, which is related to the cognitive and recognition processing of faces, is expressed in later cognitive processing stages [7,8]. Rosenfeld et al. [9] and others proposed a Complex Trial Protocol (CTP), an innovative concealed information test technique with strong countermeasure resistance. In CTP experiments, a three-stimulus protocol is usually adopted, which involves presenting the subjects with probe, irrelevant, and target stimuli. By analyzing the brainwave changes induced by these different types of stimuli and meticulously categorizing and comparing them, the CTP can be used to explore how individuals differentiate between familiar and unfamiliar faces or between their own faces and those of others. The amplitude of P300 reflects the brain’s attention to and depth of processing of the target stimulus, and the inversely proportional relationship between the amplitude of P300 and the probability of occurrence of the target stimulus reveals the brain’s sensitivity to rare or unexpected events. This sensitivity is a manifestation of the brain’s adaptability and learning ability, indicating that our cognitive system is highly focused on novel or significant events in the environment to facilitate effective information processing and decision making [10,11,12].

In research on cognitive activities using the ERP technique, in order to reduce noise and highlight the underlying neural responses, ERP epochs are averaged [13]. The purpose of averaging ERP epochs is to enhance the signal-to-noise ratio of the ERP; thereby, key features of ERP components, such as peak amplitude, total waveform area, and latency, can be extracted [14,15]. Signal processing techniques such as peak detection [16], and statistical methods such as the bootstrap method were used in ERP analysis [17]. With the development of technology, machine learning and neural network approaches have been incorporated into cognitive activity classification. These machine learning technologies have achieved significant progress that highly improved the accuracy of cognitive activity classification [18,19,20].

Eduardo et al. [21] proposed the EEG-Inception model, which incorporates a tailor-made Inception module for ERP data processing. EEG-Inception applies convolutional layers with various kernel sizes in parallel to adapt to different time scales, allowing simultaneous capture of both short-term and long-term signal patterns. This unique design enables EEG-Inception to extract complex features from EEG signals efficiently, thus significantly enhancing ERP decoding accuracy in BCI applications. Li et al. [22] introduced a novel approach combining a multi-scale convolutional neural network (CNN) with a graph-based recursive attention model (Multi-Scale Convolution with Graph-Based Recursive Attention Model, MCGRAM) for subject-independent ERP detection. By integrating multi-scale CNNs, graph convolution networks, LSTM, and self-attention mechanisms, this structure extracts significant EEG signal features, thereby improving the accuracy of ERP detection. Wang et al. [23] proposed a multi-scale EEGNet (MS-EEGNet) model to address the challenge of classifying ERPs across individuals in a BCI system of Rapid Serial Visual Presentation (RSVP). This model enhances robustness by employing parallel convolutional layers with multi-scale cores to extract discriminative information in the temporal domain. However, the fixed lengths of the multiple scales used in the above research are limited in their information capture efficiency. Excessively large scales cannot capture exact signal details, while excessively small scales lead to an overabundance of model training parameters, which affects the processing efficiency [24].

Therefore, dynamically generated scales that can learn and adjust the size according to the characteristics of temporal ERP signals would significantly enhance model performance. This approach not only precisely captures significant information in the signals but also improves model efficiency. Additionally, in ERP research, different individuals display unique ERP characteristics. These inter-individual differences, reflected in the latency, amplitude, and width of ERP waveforms, cause each person’s ERP signal to have specific personalized patterns. Therefore, the diversity of ERP features among subjects inevitably leads to significant validation errors when analyzing and interpreting ERP data. To overcome this challenge, a more flexible model that can fully adapt to each subject’s specific neural response patterns needs to be developed, thus enhancing the generalization ability when interpreting ERP data. The objective of our paper is to address a binary classification problem on an imbalanced ERP dataset related to self-face and non-self-face recognition. We assess the performance of our model using the balanced accuracy and F1 score.

The structure of this article is as follows: Section 2 describes the CTP experiment, the dataset, the structure of our model, and the performance metrics used in this paper to evaluate our model. Section 3.1 describes the statistical ERP results of our dataset. Section 3.2 and Section 3.3 detail the evaluation and comparison of model performance. Section 3.4 and Section 3.5 provide a discussion on the parameters of our model.

## 2. Materials and Methods

### 2.1. CTP Experiment and Dataset

A total of 118 participants (54 males, 64 females; age = 21.65 ± 6.21 years), including undergraduate and graduate students, were recruited for this experiment. Each participant signed an informed consent form prior to the experiment. The experimental procedure was approved by the Institutional Review Board of the Institute of Automation, Chinese Academy of Sciences.

Experimental Paradigm: Face images are widely used in psychological experiments not only to study face perception but also for research on emotion processing, social decision making, interpersonal relationships, in-group/out-group biases, and stereotyping. Subliminal presentation techniques, such as backward masking, are also employed to present faces below the level of conscious awareness. Figure 1 illustrates a single experimental trial. In a CTP test, when an image stimulus appears, participants must react by pressing the “A” key with their left hand, indicating that they have seen the image. The image is displayed for 300 ms, during which participants can press the key, followed by a random blank screen for 1300–1650 ms, requiring a response within 1500 ms. Subsequently, a numeric stimulus is displayed for 300 ms, followed by another random blank screen for 1300–1650 ms, leading to another image stimulus, followed by moving to the next trial. When the number “11111” appears, participants should respond by pressing the left arrow key with their right hand; for other numbers (“22222”, “33333”, “44444”, or “55555”), they should press the right arrow key. Participants must respond to the numeric stimulus within 1500 ms; failing to respond and incorrect responses are counted as errors. Participants rest for at least 30 s between each test block, during which they may relax but must not touch any equipment above the head. Each participant’s dataset consists of 10 blocks, each containing 60 trials with a probe-to-irrelevant stimulus ratio of 1:4, totaling 12 and 48, respectively, as well as a target-to-non-target stimulus ratio of 1:4, also totaling 12 and 48, respectively. This part comprises 10 blocks and 600 trials, with each trial lasting approximately 3000 ms, totaling about 50 min (including rest periods). Signal isolation requires the amplitude to be averaged over multiple trials. To enhance the signal-to-noise ratio, data from single trials across the 10 blocks are averaged.

### 2.2. Preprocessing

When preprocessing event-related potential (ERP) data using the EEGLAB software package in MATLAB R2019b, we begin by importing the position information of electrodes in the order of participant numbers and marking or modifying specific electrodes according to the standard electrode position template (BESA standard). Unwanted electrooculography (EOG) leads, including ‘EOG’, ‘AF7’, and ‘AF8’, are removed to minimize artifacts such as eye movements that could affect the data. The data undergo bandpass filtering to eliminate high-frequency noise and low-frequency drift, with settings of 30 Hz for the low-pass band and 0.1 Hz for the high-pass band. The data are then re-referenced to the bilateral mastoids to enhance the stability and comparability of the recordings. Specific event trials are extracted, and baseline corrections are performed to negate pre-experiment influences. Bad leads are automatically identified and excluded, with interpolation being used to repair these leads. Trials with excessive noise are automatically identified and removed to reduce the impact of outliers. Independent Component Analysis (ICA) is conducted, followed by the use of the MARA algorithm to automatically identify and remove artifact components related to eye movements. Finally, specific event types are selected based on different stimulus categories, individual trials are extracted, and ERP signals are averaged for each event. The sampling rate for the preprocessing is 1000 Hz.

### 2.3. Experimental Sample

To facilitate cross-subject experiments, the smallest unit for training and testing dataset division was defined as an individual participant’s ability to recognize five different facial categories. All experiments were conducted on the following hardware configuration: an Intel Core i5-13600 CPU at 3.50 GHz, 16 GB RAM, and an NVIDIA RTX A6000 GPU. In our experiment, the training of the model involved meticulous hyperparameter tuning to maximize performance. The learning rate was set to 1 $\times \;10^{-5}$, and the weight decay was set to 1 $\times \;10^{-4}$ to prevent overfitting and enhance the model’s generalizability. The range of hyperparameter λ in Equation (Equation 6) was set to 3, 4, 5, 6, 7, 8, 10, with λ = 5 showing the best performance. To improve the model’s robustness against different classes, a weighted mechanism was implemented in the BCEWithLogitsLoss function to address the imbalance between positive and negative samples [25]. Additionally, to further enhance the model’s generalizability, a dropout rate of 0.5 was used to randomly drop network connections during training, reducing the model’s reliance on training data. The model’s output was determined by a threshold, with predictions over 0.5 being classified as positive samples and those under 0.5 being classified as negative samples. The data division strategy split the dataset into training and testing sets in a 9:1 ratio, ensuring sufficient data for learning during the training phase and effective performance validation during the testing phase. Finally, to ensure robust evaluation results, three-fold cross-validation was employed, further ensuring the reliability of the assessment and the model’s capabilities for generalization to new data.

### 2.4. A Multi-Scale Model Based on ERP-Generated Weight Masks

Different individuals, as well as the same individual under varying conditions, may exhibit distinct temporal characteristics in their ERP signals, such as changes in peak timing and waveform duration [26,27]. ERP signals can be influenced by multiple factors during experimental procedures, displaying dynamically changing characteristics. A fixed window length may struggle to adapt to these changes, potentially leading to the loss of crucial information or interference from irrelevant data. In response to the issue of dynamically adjusting time windows, Qian et al. [28] proposed a novel deep learning model—the Dynamic Multi-Scale Convolutional Neural Network. This model employs a variable-length filter generator to dynamically determine the optimal filter length for time-series samples, thereby capturing features across different time scales. This approach does not rely on prior knowledge to set filter scales but instead learns from the data to extract specific multi-scale feature representations for each sample. Compared to traditional methods, this strategy enables more precise capture of critical information in time-series signals, thereby enhancing the classification accuracy and the generalization ability of the model. To address the limitations of fixed time window lengths in ERP signal processing and the effects of individual differences, this paper builds upon the filter generator model proposed by Qian and others. It employs a Multi-Scale Variable One-Dimensional Convolutional Kernel Length Convolutional Neural Network to extract ERP component features, allowing the deep learning network to better adapt to the variability and complexity of EEG data. By generating convolutional kernels of varying lengths based on the differences among subjects, the network dynamically captures critical information from ERPs, enhancing the model’s ability to recognize signals under different individual and conditional circumstances. Through this refined and focused processing strategy, we can gain a deeper understanding of the functions of various ERP components and provide a more detailed method of analysis for ERP research. In summary, by dynamically adjusting the filter length using a 0–1 masking mechanism, where the specific length is related to the characteristics of the ERP signal, the model can simultaneously capture features at different time scales, such as the P300 and N170 components with different waveform durations.

#### 2.4.1. Data Input

The input of the model consists of a series of time-series data, with a primary focus on the time segments corresponding to ERP (event-related potential) components. The task involves extracting relevant features from the given segments of the signal. These extracted features are then utilized to generate the appropriate length of the convolutional kernels. The ERP signal can be described as follows:(1)$$S=\left[\begin{array}{cccc} S_{11} & S_{12} & \cdots & S_{1T}\\ S_{21} & S_{22} & \cdots & S_{2T}\\ \vdots & \vdots & \ddots & \vdots \\ S_{C1} & S_{C2} & \cdots & S_{CT} \end{array}\right]$$

In this context, $S_{CT}$ represents the signal strength of the *C*-th channel at the *T*-th time point of the input signal.

#### 2.4.2. Variable-Length Convolutional Kernels

During the convolutional kernel length construction phase, this model employs a series of convolutional layers and introduces the core design of variable-length convolutional kernels. Compared to traditional fixed-size one-dimensional convolutional kernels, the essence of this design lies in its flexibility; by generating a 0–1 mask and multiplying it with the weights of the convolutional kernels of the same dimension, the model can adaptively capture features across various time scales. The generation of the mask is based on the output of two fully connected layers, which integrate and transform previous features with the embedded representations of the convolutional kernels, thereby producing a mask vector for each time step. First, the data are expanded and convoluted with the filter $W_{i}^{F}$ on *S* to obtain the signal characteristics $e_{i}\in R^{C\times T}$, as defined in Equation (Equation 2). Here, $W_{i}^{F}$ denotes the *i*-th filter, with a convolution kernel size of (1,3) and a total of *K* kernels. The collective results of all convolutions are denoted as *E*.
(2)$$e_{i}=W_{i}^{F}\cdot S+b_{i}$$
(3)$$E={\left\{e_{1},\;e_{2},\;\cdots ,\;e_{K}\right\}}^{T}$$

In order to generate a weight mask value for each sample based on its temporal waveform characteristics, a global average pooling is first conducted along the dimension of EEG channels, reducing the channel dimension C to 1, $E^{\prime }\in R^{K\times 1\times T}$.
(4)$$E^{\prime }=GAP\left(E\right)$$

Subsequently, in the analysis of event-related potential (ERP) data, a specific segment of the ERP data is initially extracted to serve as the input $L\in R^{K\times 1\times l}$ for generating mask values, with a length denoted as *l*. This segment starts at point P and ends at P+l.
(5)L=T[P:P+l]⊆T

To establish a variable dynamic convolution baseline, *K* one-dimensional convolution kernels matching the dimensions of the extracted data are randomly initialized, and a fixed convolution kernel weight $W\in R^{K\times 1\times l}$ is randomly generated according to a Gaussian distribution, as shown in the orange section of Figure 2. Each convolution kernel is set to a fixed length *l*, and the fixed one-dimensional convolution kernel weights are used as a baseline. Then, the *K* segments of ERP data are concatenated with the *K* one-dimensional convolution kernels $W_{i}$ in the time domain. The methods proposed by Qian et al. [28] are as follows: During the mask generation phase, the concatenated input data $R^{K\times 1\times 2l}$ are processed through two fully connected layers. The first fully connected layer has biases of $W_{m1}$ and $b_{m1}$, respectively, and a size ratio set to *2l*:*l*, with the aim of reducing the data dimensions. The task of the second fully connected layer is to output a specific value, with a size ratio set to *l*:1, and the weights and biases are designated as $W_{m2}$ and $b_{m2}$, respectively. To constrain this value between 0 and 1, a sigmoid (σ) activation function is incorporated after the second layer to produce a decimal value $r_{i}$. This process is represented as follows:(6)$$r_{i}=\sigma \left(W^{m2}\left(W^{m1}\cdot \left(L_{i}⊕W_{i}\right)+b^{m1}\right)+b^{m2}\right)$$

This value $r_{i}$ is multiplied by a scale size to output an ideal mask length value $l_{i}$ ranging from 0 to 1.
(7)$$l_{i}=r_{i}\times l$$

To ensure that this process remains differentiable during training, the obtained length value $l_{i}$ is incorporated into the carefully designed differentiable Formula (Equation 8). In this formula, *i* represents the scale index, *j* ranges from 1 to the time step *l*, σ denotes the sigmoid activation function, and λ stands for the sharpness parameter.
(8)$$\forall j\in 1,\;2,\;\cdots ,\;l,\;m_{ij}=\sigma \left(\lambda \left(l_{i}-j\right)\right)$$

Using the aforementioned Formula (Equation 8) [29], a 0–1 soft mask vector is generated for each time step. The collection of all masks is denoted as follows:(9)$$M={\left\{m_{1\times l},\;m_{2\times l}\cdots ,\;m_{K\times l}\right\}}^{T}$$

#### 2.4.3. Convolution Kernel Application

The generated mask values $M\in R^{K\times l}$ are multiplied by the fixed convolution kernel weights that were initially generated according to a Gaussian distribution, thereby producing variable one-dimensional convolution kernels $W\in R^{K\times l}$. These variable one-dimensional convolution kernels possess a structure that matches the original ERP data (i.e., the number of channels multiplied by the number of time points), allowing the length of each kernel to directly correspond to the dimensions of specific sample data.

By convolving these generated one-dimensional convolution kernels with the corresponding ERP data, we are able to extract unique features for each sample. Formula (Equation 10) adjusts the length of the convolution kernels using the mask values, producing the weight module inside the dashed box shown in Figure 2.
(10)$$W^{\nu }=W⊗M$$
$$W^{\nu }=\left\{W_{1}^{\nu },W_{2}^{\nu },\cdots ,W_{K}^{\nu }\right\},W_{i}^{\nu }\in R^{l\times 1}$$

The convolution layer now features variable-length filters that better adapt to the specific characteristics of the input data. The use of masks to generate variable-length filters means that the last portion of each filter contains some zeros, which could impede the convolution of the time series. To address this issue, we pad the input sequence to obtain a length $S_{i}^{\prime }\in R^{(L+l)\times 1}$, which is represented as follows:(11)$$S_{i}^{\prime }=\left[\begin{array}{ccccc} S_{11} & S_{12} & \cdots & S_{1T} & 0\\ S_{21} & S_{22} & \cdots & S_{2T} & 0\\ \vdots & \vdots & \ddots & \vdots & \vdots \\ S_{C1} & S_{C2} & \cdots & S_{CT} & 0 \end{array}\right]$$The dataset can be represented as follows:(12)$$S^{\prime }=\left\{S_{1}^{\prime },S_{2}^{\prime },\cdots ,S_{K}^{\prime }\right\},S_{i}^{\prime }\in R^{(L+l)\times 1}$$

Convolution is performed on the length S′ obtained from padding using a convolution layer with added masks and channel-wise convolution [30]. For the *i*-th feature map data $S_{i}^{\nu }$, the *i*-th convolutional kernel is used to extract features from the *j*-th EEG channel. The obtained features are concatenated together. This is represented as follows:(13)$$D=\sum_{i=1}^{k}\sum_{j=1}^{C}W_{i}^{\nu }\times S_{ij}^{\prime }+b_{i}^{\nu }$$
$$D\in R^{K\times C\times T}$$

#### 2.4.4. Classification Layer

After the depthwise separable convolution layers, a batch normalization (BN) layer and a dropout layer are used to prevent gradient explosion and avoid overfitting. In the final module, a linear fully connected layer is used as the output layer to perform a binary classification.
(14)Y=WD+b

### 2.5. Metric Indicators

To thoroughly assess the performance of the proposed method in the classification task of ERP signals, this study specifically addresses the data label imbalance issue in binary classification scenarios. The classification task aims to distinguish between positive samples (the subject’s own face) and negative samples (faces of strangers), where the ratio of positive to negative samples is 1:4. This imbalance poses additional challenges in evaluating classification performance. For this reason, this study employs the following five metric indicators: balanced accuracy (BA), F1 score, true positive rate (TPR), true negative rate (TNR), and area under the curve (AUC). These metrics comprehensively reflect the model’s ability to handle imbalanced data categories.

Among the metrics, TP represents the number of positive samples correctly predicted as positive by the model. TN indicates the number of negative samples correctly predicted as negative. FP refers to negative samples incorrectly predicted as positive, and FN denotes positive samples incorrectly predicted as negative. In the context of an imbalanced dataset, BA is chosen as the measurement criterion because it equally considers the classification performance of all categories, even when some categories have fewer samples. It calculates the average of the TPR and TNR, ensuring that the evaluation results are not biased toward the majority class, thereby providing a fair and comprehensive method of performance measurement. The F1 score, as the harmonic mean of precision and recall, is another key indicator for assessing model performance in category imbalance. It maintains a high true positive rate while also considering the false positive rate, ensuring that the model does not sacrifice the recognition of minority classes in favor of the majority.
$$\begin{array}{cc} \mathrm{BA} & =\frac{\mathrm{TPR}+\mathrm{TNR}}{2}\\ \mathrm{TPR} & =\mathrm{Recall}=\frac{\mathrm{TP}}{\mathrm{TP}+\mathrm{FN}}\\ \mathrm{TNR} & =\frac{\mathrm{TN}}{\mathrm{FP}+\mathrm{TN}}\\ \mathrm{FPR} & =\frac{\mathrm{FP}}{\mathrm{FP}+\mathrm{TN}}\\ F1 & =2\times \frac{\mathrm{Recall}\times \mathrm{Precision}}{\mathrm{Recall}+\mathrm{Precision}}\\ \mathrm{Precision} & =\frac{\mathrm{TP}}{\mathrm{TP}+\mathrm{FP}} \end{array}$$

The AUC is an important performance metric, especially when dealing with imbalanced data, as it provides an accurate measure of performance. In this study, particular attention is paid to the AUC metric to ensure the accuracy and robustness of our proposed method when facing an imbalanced dataset with a ratio of positive to negative samples of 1:4.

## 3. Results

### 3.1. Statistical ERP Results

As illustrated in Figure 3, the overall averaged ERP signal at the Pz channel (*n* = 118) is shown, where the ERP waveform induced by self-face stimuli is depicted by a blue solid line, and that induced by stranger-face stimuli is depicted by a red dashed line. The solid and dashed lines represent the mean ERP values, while the colored shadow areas indicate the ±1 standard deviation range. The gray shaded areas denote time windows where significant differences exist between conditions (*p* < 0.05). In the graph, unlike with stranger-face conditions, the self-face stimuli elicit a P300 component between 300 to 500 ms, peaking around 400 ms. The N200 component was not observed under the self-face condition. Under stranger-face conditions, the P300 component was not observed, while the N200 component was elicited between 180 to 250 ms, with a peak latency at 211 ms.

A *t*-test comparing the average ERP amplitudes under familiar-face and stranger-face conditions revealed significant differences within the 189–250 ms and 311–527 ms time windows, corresponding to the N200 and P300 components, respectively. Compared to stranger-face-induced ERPs, familiar faces induced a larger P300 wave, whereas stranger-face stimuli elicited a larger N200 wave. These phenomena suggest that the N200 and P300 components may represent the brain’s differing responses to cognitively processing familiar and unfamiliar faces. It was shown that significant differences from 600 ms to 1000 ms indicate late ERP waveforms, which may reflect access to and retrieval of face-related information from long-term memory [31,32].

### 3.2. Face Recognition Results

Our result includes the performance of EEGNet, PLNet, EEG-Inception, DeepConv, and a multi-scale model that we proposed based on ERP-generated weight masks. The performance evaluation metrics include the BA, F1 score, TPR, FPR, and AUC. According to the model performance results shown in Table 1, our model achieved significant accomplishments across various performance evaluations. Particularly for the BA and AUC metrics, our approach demonstrates superior performance compared to the other methods. Specifically, our model achieves a BA of 0.9320, outperforming EEGNet (0.8589), PLNet (0.8308), EEG-Inception (0.8829), and DeepConv (0.7735), showing the accuracy and efficiency of our approach. In terms of the AUC metric, our model scores an impressive 0.9676, indicating superior discriminative ability compared to the other methods. This result is particularly notable in terms of classification accuracy, especially compared to DeepConv, where our method improves the AUC by approximately 0.1199, and it exhibits lower variability, reflecting its exceptional stability and consistency. Regarding the F1 score, our model leads with a score of 0.8854, significantly ahead of DeepConv’s 0.6406, indicating a greater combined effect of precision and recall.

Regarding the TPR, our model’s performance surpasses that of PLNet (0.7215) and DeepConv (0.6991) but is lower than that of EEGNet (0.7624) and EEG-Inception (0.7436). The higher TPR indicates that the model is better at correctly identifying true positives (the subject’s own face). As for the FPR, our model performed worse than EEGNet (0.1940) and EEG-Inception (0.1697) but improved upon PLNet (0.2089) and DeepConv (0.2684). The lower FPR suggests that the model is more effective at avoiding mistakenly recognizing non-subject faces as subject faces, thus reducing the number of false alarms. In face recognition tasks based on ERP signals, a lower FPR is particularly crucial, as it diminishes the likelihood of misidentifying non-target faces as target faces, thereby reducing potential confusion or misoperations caused by incorrect judgments.

### 3.3. Loss Comparison

To assess the effectiveness of our model, we conducted performance tests on the validation set and compared the results with those of current state-of-the-art models, including EEGNet, EEG-Inception, and PLNet. The validation loss (val-loss) is a significant indicator of a model’s generalization ability, and, as shown in the figures, the validation losses of the different models exhibited significant differences as the training epochs increased.

Throughout the testing process, our model (labeled as “ours”) demonstrated the lowest validation loss, indicating higher predictive accuracy and superior generalization capabilities on out-of-sample data. Specifically, as illustrated in Figure 4, our model rapidly reduced the validation loss in the early stages of training and stabilized at a low loss value after approximately 25 epochs, confirming its quick learning ability. In the later stages of training, although the loss values of all models tended to stabilize, our model still maintained the lead. In contrast, other benchmark models, particularly EEG-Inception, showed higher validation losses throughout the training process, which might suggest an overfitting to the training data. Moreover, although EEGNet and PLNet performed better, they still could not match the performance of our model. This result clearly indicates that our model, when dealing with problems in this validation set, is more effective in reducing overfitting and provides more accurate predictions compared to other advanced technologies.

### 3.4. Comparison of the Standard Model and Masked Weight Model

The masked weight model employs the generation of 0–1 masks, which are multiplied by the weights of convolution kernels of the same dimension. Our model can adaptively capture features across various time scales. The weights generated by the convolution are focused on capturing local patterns on smaller scales, such as the N200 and P300 components, which are more critical for the classification task. When the length of the mask values, i.e., the length of the weights produced by multiplication, aligns with these local features, incorporating masks tends to yield better results. The standard model primarily consists of two convolutional layers and one linear layer, whereas the masked weight model introduces a weight mask on this basis. This work covers five significant performance indicators, spanning four scale factors (5, 10, 20, 30), which represent the multi-scale K levels used in this work.

The comparison of the FPR across different scales indicated that, although both models maintained relatively consistent performance, the inclusion of the masked weight model significantly reduced the FPR at scales K = 10 and K = 20 by 6.95 and 5.33 percentage points, respectively (Figure 5). The TPR metrics show that at scales of 5, 10, and 20, the performance of the masked weight model surpassed that of the standard model, with very minor variations in performance between the two models (Figure 6). The BA results highlight the effectiveness of the masked weight model relative to the standard model, showing slightly higher accuracy across all scales. Specifically, at a scale of K = 5, the BA of the masked weight model reached 0.8784, compared to 0.8420 for the standard model, demonstrating an improvement of 3.64 percentage points. As the scale increased to K = 10, the BA of the masked weight model further increased to 0.9211, which was significantly higher than the standard model’s 0.8298 by 9.13 percentage points. At larger scales of K = 20 and K = 30, the BA values for the masked weight model were 0.9212 and 0.9315, respectively, compared to 0.8376 and 0.8601 for the standard model, showing improvements of 8.36 and 7.14 percentage points, respectively (Figure 7).

The F1 score, which combines precision and recall into a single metric, indicates that both models performed well, with the masked weight model having a slight advantage. Notably, at scales of K = 10 and K = 20, the F1 scores for the masked weight model were 0.8710 and 0.8711, significantly higher than the standard model’s 0.7265 and 0.7411, respectively (Figure 8). Finally, the AUC provides a comprehensive measure of model performance across all classification thresholds. The masked weight model achieved high AUC values compared to the standard model, indicating its strong discriminative power in classifying ERP signals (Figure 9). The error bars represent standard deviations, highlighting the variability in each model’s performance. In summary, the inclusion of the masked weight model demonstrates significant improvements across all considered performance metrics. These results emphasize the importance of incorporating weight masks, which not only enhance the model’s learning capabilities but also improve its generalization over data.

### 3.5. The Impact of the Convolutional Scale

We explore the effects of scale length (filter size) and feature extraction locations [*P*, *P* + *l*] during the mask generation process on the performance of convolutional neural networks. A uniform scale (K) of 30 is set, and the filter lengths are adjusted to 100, 200, 300, and 600 to evaluate the impact of using different reference values for mask generation at various ERP time points [*P*, *P* + *l*]. In the case of ERP signal classification, using the AUC metric helps assess the model’s generalization capabilities across different potential ERP waveforms. This is crucial for ensuring the model’s practicality in practical applications, especially when ERP waveform features may vary significantly. Therefore, we evaluate this impact using only the AUC metric. The experimental results show distinct differences in AUC performance based on the reference values selected at different ERP window positions. When extracting features with a fixed convolutional kernel count of 30, we observe that the model’s AUC values varied at different ERP time points as the window length increased. As illustrated in Figure 9, when the window length is set to 300, the AUC value shows a consistent trend of improvement with greater positional information, capturing more useful time-series features within the [600, 900] feature position. However, when the window lengths are set to 100 and 600, the AUC values exhibit different trend characteristics. The results for a length of 100 indicate a peak performance in the [300, 400] region for the ERP window position, followed by a slight decline in performance in subsequent areas due to the window length capturing irrelevant information. At a scale length of 200, the performance peaks in the ERP window [400, 600] and then sharply declines. Similarly to the scale of 100, the experiments with a scale of 600 also show non-monotonic performance variations with ERP window position changes. These results emphasize the importance of considering appropriate window lengths and corresponding positional information in designing convolutional neural networks (Figure 10).

Our findings indicate that the selection of window positions at different scale sizes significantly affects the efficiency of feature extraction from time-series data and ultimately model performance. After a detailed comparative analysis regarding the ERP classification problem in this study, we find that at a window length of 200, the model exhibits optimal performance in the ERP feature position [*P*, *P* + *l*] with the interval [400, 600]. This peak suggests that under this parameter setting, the model captures the most relevant and distinctive features within that time window. By analyzing the AUC performance at different scale sizes, we observe that the model’s performance is optimal between 200 to 600 ms, corresponding to the time segments of the N200 and P300 components, which are typically associated with key event processing in cognitive tasks. This signal segment is particularly important for recognizing and classifying ERP signals. The experimental results show that when the convolutional neural network’s filter length is adjusted to 200 and feature positions are selected within this ERP time segment, the model’s AUC value reaches its highest peak. This indicates that the features extracted within this time range are the most distinctive and relevant for the model, enabling a more accurate classification of facial ERP signals.

## 4. Conclusions

In this paper, we proposed a dynamic neural network model for CTP-based face recognition. The results from the 118 subjects indicate that, compared to other methods, our model achieves a balanced accuracy (BA) of 97.67%, a TPR of 65.43%, an FPR of 25.97%, an AUC of 96.76%, and an F1 score of 88.54%. Our model employs soft masks to generate dynamic and multi-scale lengths, making efficient use of data signal features. Compared to other promising technologies addressing inter-subject variability, our framework dynamically adapts to new participants during the inference process. Traditional multi-scale methods require the manual setting of different filter scales while relying on domain knowledge, and they may not be suitable for all datasets. In contrast, the dynamic scale generation method does not require these manual adjustments.

Despite the positive outcomes of this study, there are still some limitations that need to be addressed in the future. Our model leads to a large and redundant number of parameters, thereby increasing the parameter complexity and time cost. Furthermore, the significant class imbalance notably affected the FPR and TPR metrics, resulting in decreased performance. It is especially noteworthy that although the model with 30 convolutional kernels ultimately achieved a higher balanced accuracy, its performance improvement tended to stabilize after 100 training epochs, suggesting a potential performance bottleneck. This phenomenon reminds us to carefully consider the balance between model capacity and computational efficiency in practice. While increasing the number of convolutional kernels can enhance model performance, it also correspondingly increases the computational load and required training time.

We also considered adopting pointwise convolutions following channel-wise convolutions, thereby constructing a lightweight model based on depthwise separable convolutions within the convolutional module. However, this approach could potentially reduce data precision for our specific dataset; therefore, pointwise convolutions were not incorporated. Concerns about overfitting led us to implement a dropout rate of 0.5. Additionally, to validate the generalization ability of our model, we utilized an extra validation set. By assessing the validation set loss, we demonstrated that our model’s generalization capability surpasses that of other SOTA models.

Future work should focus on several key areas to enhance the performance and applicability of EEG models. First, by dynamically adjusting the filter length, the model can simultaneously capture features at different time scales. For example, in the experimental paradigm of the CTP, the model can be adapted to extract the relevant time segments for the P300 and N170 components. For other experimental paradigms, the model can be adjusted to extract different components based on the specific components elicited by the paradigm. Second, it is essential to collect and utilize larger and more diverse datasets that encompass a wide range of participants and cognitive tasks [36,37]. Such datasets will enhance the model’s generalization ability, allowing it to perform robustly across various scenarios. Finally, improving the existing model by adapting the entire model structure to lightweight networks, such as depthwise separable convolutions or mixed convolutions, will help reduce computational complexity [38,39]. These adaptations can lead to more efficient models that maintain high performance while being computationally feasible for real-time applications and deployment on resource-constrained devices.

## Figures and Tables

**Figure 1 sensors-24-04368-f001:**
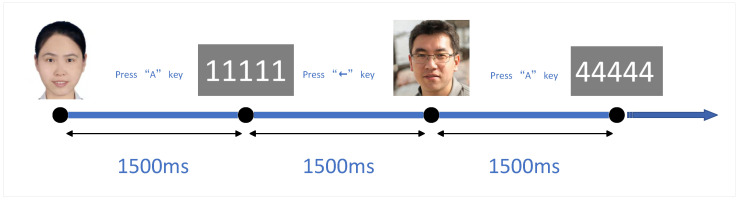
Flowchart of a single experiment. Note: The faces in the flowchart are generated by AI. The experimental paradigm and procedure used are the same as those in [25].

**Figure 2 sensors-24-04368-f002:**
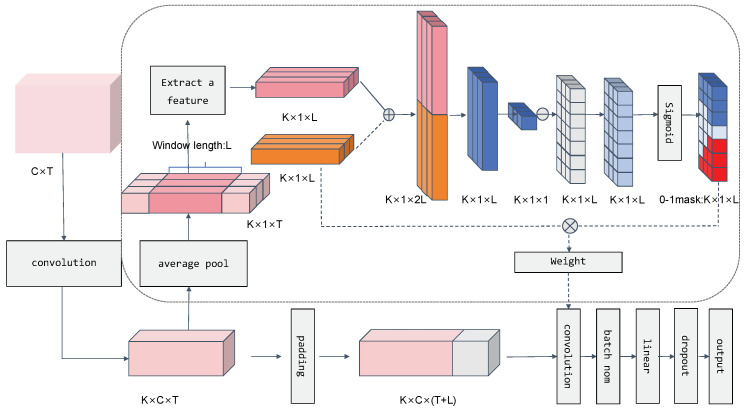
Multi–scale variable–length depthwise separable convolution. In the 0–1 mask, the blue cubes represent a mask value of 1, the red cubes represent a mask value of 0, and the light blue represents values between 0 and 1. ⊕ represents concatenation, ⊗ represents multiplication, and ⊖ indicates the use of Formula (Equation 8) on the next gray module to obtain the light blue module, where *i* is the index of the scale, and *j* is the time step from 1 onwards. Solid lines represent the flow of ERP data, while dashed lines indicate the flow of weights. The gray dashed box encompasses the dynamic generative weight mask model, while outside the dashed box is the standard model.

**Figure 3 sensors-24-04368-f003:**
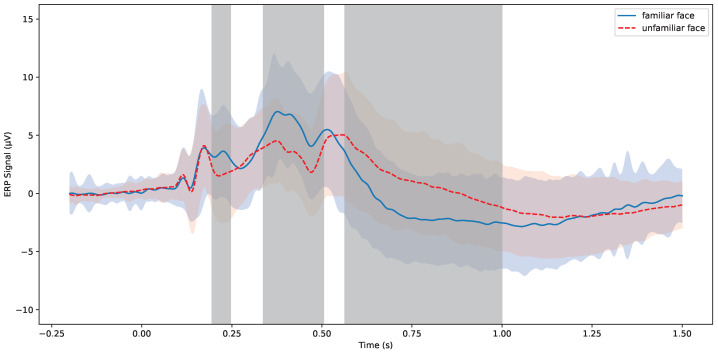
The average value of the Pz channel. The blue portion represents the standard deviation of positive samples, while the red region represents the standard deviation of the averaged negative samples. The gray shadow indicates the region of significant difference between familiarity and unfamiliarity (*p* < 0.05).

**Figure 4 sensors-24-04368-f004:**
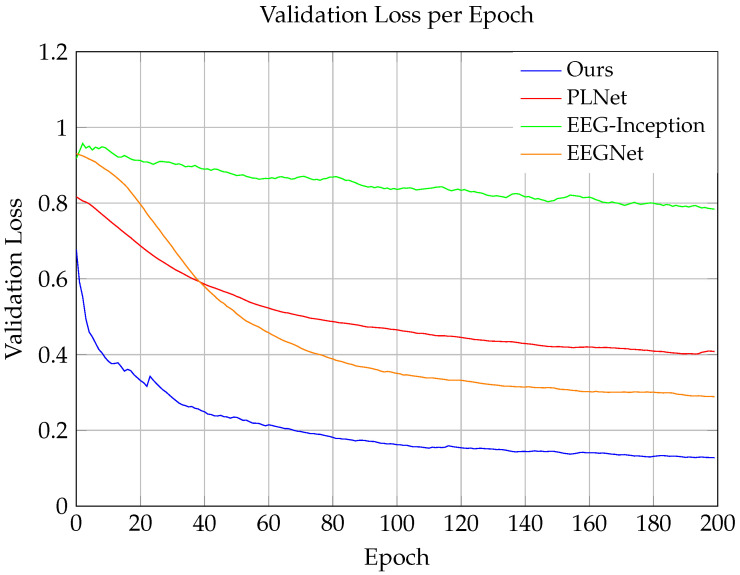
Comparison of the validation losses across models.

**Figure 5 sensors-24-04368-f005:**
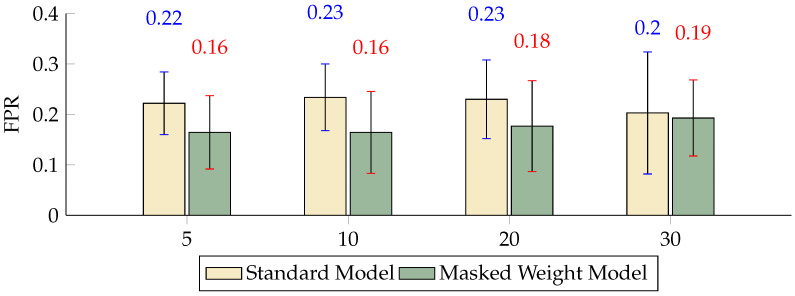
Differences in the FPR between the masked weight model and the standard model with different numbers of scales (*p* < 0.01).

**Figure 6 sensors-24-04368-f006:**
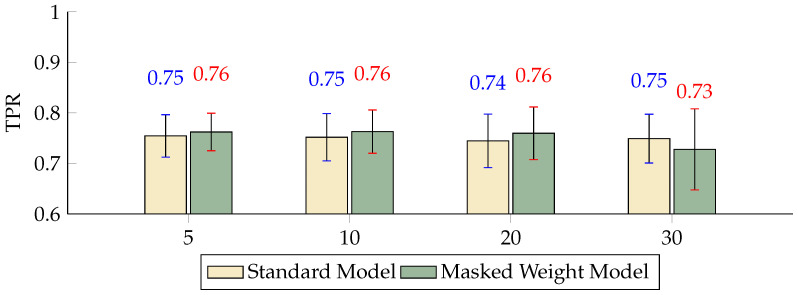
Differences in the TPR between the masked weight model and the standard model with different numbers of scales (*p* < 0.5).

**Figure 7 sensors-24-04368-f007:**
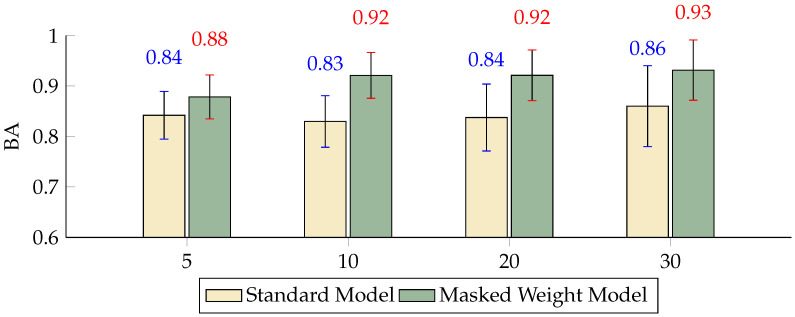
Differences in the BA between the masked weight model and the standard model with different numbers of scales (*p* < 0.01).

**Figure 8 sensors-24-04368-f008:**
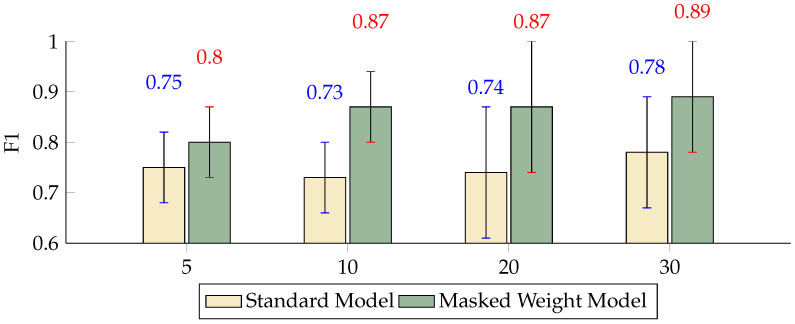
Differences in the F1 score between the masked weight model and the standard model with different numbers of scales (*p* < 0.01).

**Figure 9 sensors-24-04368-f009:**
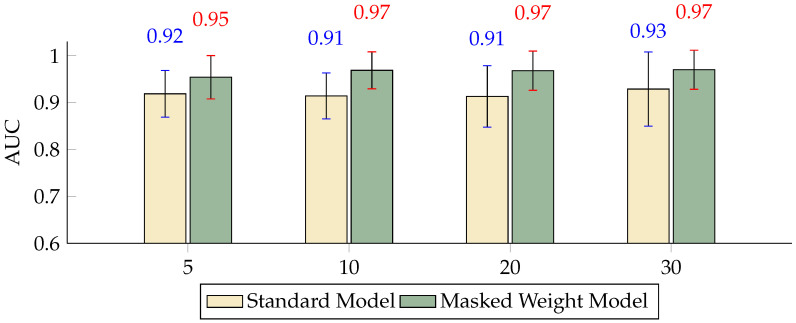
Differences in the AUC between the masked weight model and the standard model with different numbers of scales (*p* < 0.01).

**Figure 10 sensors-24-04368-f010:**
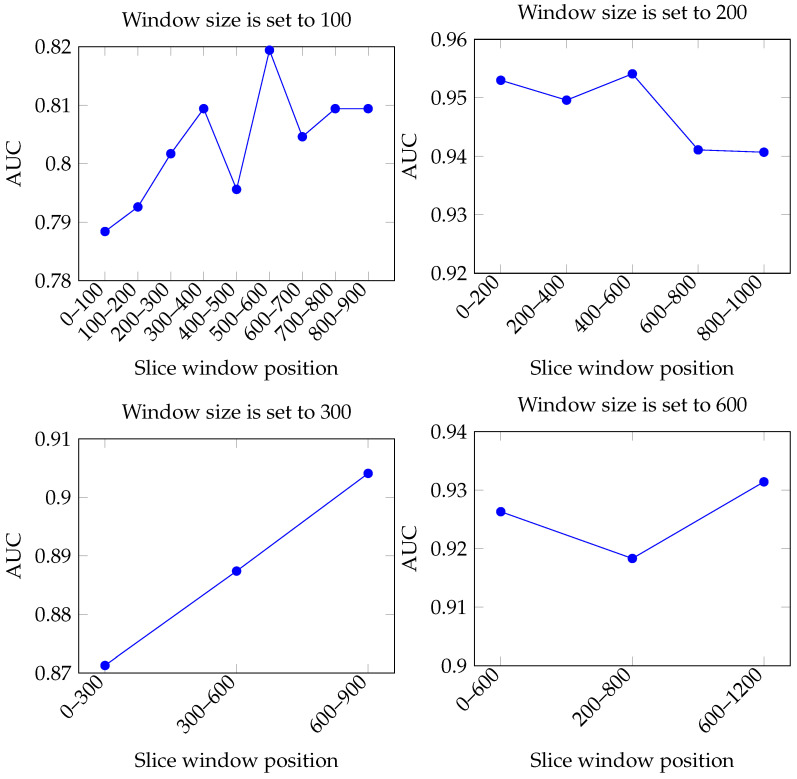
The effect of the window size and window position.

**Table 1 sensors-24-04368-t001:** Comparison of the performance of different methods.

Method	BA↑	TPR↑	FPR↓	F1↑	AUC↑
EEGNet [33]	0.8589±0.0721	0.7624±0.0485	0.1940±0.0623	0.7835±0.1269	0.9401±0.0530
PLNet [34]	0.8308±0.0525	0.7215±0.0437	0.2089±0.0694	0.7164±0.1036	0.9118±0.0555
EEG-Inception [21]	0.8829±0.0498	0.7436±0.0384	0.1697±0.0566	0.8043±0.0822	0.9489±0.0425
DeepConv [35]	0.7735±0.1120	0.6991±0.0930	0.2684±0.1549	0.6406±0.1813	0.8477±0.1044
ours	0.9320±0.0655	0.7257±0.0734	0.2017±0.0778	0.8854±0.1026	0.9676±0.0444

Note: The symbol “↑” indicates that higher values signify better model performance. Conversely, the symbol “↓” denotes that lower values indicate better model performance.In the performance metrics, the bold data represent the best performance indicators for the same method.

## Data Availability

The raw data supporting the conclusions of this article will be made available by the authors on request.

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
