# Peer review of "A Dynamic Multi-Scale Convolution Model for Face Recognition Using Event-Related Potentials"

_sensors, 2024, doi:10.3390/s24134368_

Round 1
Reviewer 1 Report
Comments and Suggestions for Authors
The paper describes an interesting study concerning recognition of human perception state from EEG data. A set of methods is selected and ordered in a workflow. The resulting method is tested against a real database and SOTA competitors. The results are encouraging. The language of the paper is clear, the logic is understandable.
There is the following drawback. The problem statement is not described properly. The paper starts with introduction (containing literature review) followed by 'Materials and methods', where good description of details is given. However, it is not quite clear what concrete problem is being solved until line 263, where it is mentioned in the context of quality metric. The problem statement (what task is being solved) should be presented explicitly immediately after introduction.
After correcting this item the paper can be published.
Author Response
Q1:
There are the following deficiencies. The problem statement is not correctly described. The paper starts with an introduction (including a literature review) and then moves to "Materials and Methods," where details are well described. However, it is only after line 263, in the context of quality metrics, that the specific problem being addressed becomes fully clear. The problem statement (the task being addressed) should be clearly stated immediately after the introduction.
Reply:
Thank you very much for your suggestion. We agree that the problem statement should be clearly presented early in the manuscript. Therefore, we have added the following sentence at line 83 in the Introduction section:
"The objective of our paper is to address a binary classification problem on an imbalanced ERP dataset related to self-face and non-self-face recognition. We assess the performance of our model using balanced accuracy and F1 score."
Reviewer 2 Report
Comments and Suggestions for Authors
The paper presents a dynamic multi-scale convolutional neural network model for recognizing familiar and unfamiliar faces using event-related potentials (ERP) data. The proposed model utilizes generated weight masks to adaptively capture features across various time scales, addressing the challenges of understanding the relationship between ERP components and facial recognition. The model demonstrates significant performance improvements over state-of-the-art models with a balanced accuracy rate of 93.20% and an F1 score of 88.54%. The research highlights the importance of dynamically adjusting time windows and convolutional kernel lengths to enhance model performance.
Stimulating Questions
-
Methodological Adaptations: How does the proposed dynamic multi-scale convolutional model improve the capture of ERP features compared to traditional fixed-scale convolutional models? What are the key advantages and potential limitations of using dynamically generated scales?
-
Performance Metrics: The paper reports significant improvements in balanced accuracy and F1 score. What specific aspects of the model's architecture contribute to these improvements, and how do these metrics compare to other state-of-the-art models in face recognition tasks?
-
Application and Future Research: Given the model's capability to adapt to new participants during the inference process, how could this approach be extended to other types of cognitive recognition tasks? What additional research is needed to further enhance the generalization ability and computational efficiency of the model?
Author Response
Q1:
Methodology Adjustment: How does the proposed dynamic multi-scale convolution model improve the capture of ERP features compared to traditional fixed-scale convolution models?
Reply:
First, K convolutional kernels are used to upsample the original data, expanding it into K dimensions. Then, a 0-1 masking mechanism dynamically adjusts the lengths of the K generated filters, with the specific lengths corresponding to the characteristics of the ERP signals. This model can simultaneously capture features at K different temporal scales.
To address how the dynamic multi-scale convolution model enhances ERP feature capture, We have expanded the content in the main text, lines 184-187.
Q2:
What are the main advantages and potential limitations of using dynamically generated scales?
Reply:
Thank you very much for your insightful question. Here are the main advantages and potential limitations of using dynamically generated scales:
Main Advantages:
Compared to other promising technologies addressing inter-subject variability, our framework dynamically adapts to new participants during the inference process.
.Traditional multi-scale methods require manual setting of different filter scales, relying on domain knowledge and may not be suitable for all datasets. In contrast, the dynamic scale generation method does not require these manual adjustments.
We have expanded the advantages in the main text, lines 441-446.
Potential Limitations:
- Dynamic scale generation requires additional computational resources to generate and adjust the filter length, increasing the model's computational complexity and training time.
- While considered for reducing data precision issues, pointwise convolutions were not incorporated due to potential precision reduction specific to the dataset.
The following points summarize potential limitations of using dynamically generated scales, which we have also added to the conclusion section, lines 447-465.
Q3:
Performance Metrics: The paper reports significant improvements in balanced accuracy and F1 score. Which specific aspects of the model architecture contribute to these improvements.
Reply:
Thank you for your question. The following aspects of our model architecture contribute to the improvements in balanced accuracy and F1 score:
The variable-length filter generator learns a soft mask to control the filter length, allowing the model to automatically adjust the filter length during training to adapt to different sample characteristics. Global average pooling layers across channels are used to select the most discriminative local patterns by retaining the most important features in each time series.
To address this question, we conducted ablation experiments to compare the performance of the dynamic model architecture with a standard non-dynamic architecture. The results of these experiments are detailed in Section 3.4, "Comparison of Standard Model and Masked Weight Model." The performance metrics of dynamic model structures significantly outperform those of non-dynamic structures. For instance, at scales of 20 and 30, the F1 scores for the dynamic model with mask weights are 0.8710 and 0.8711, respectively, compared to 0.7265 and 0.7411 for the standard model. Similarly, the BA values for the dynamic model are 0.9212 and 0.9315, respectively, whereas the standard model's BA values are 0.8376 and 0.8601.We have added explanations in lines 358-366 of the main text to explain how using a dynamic model architecture improves model performance.
Q4:
How do these metrics compare to other state-of-the-art models in the face recognition task?
Reply:
A comparison of the performance of other state-of-the-art models in face recognition tasks can be found in Table 1. For the balanced accuracy metric, our model achieved 93.20%, outperforming EEGNet (0.8589), PLNet (0.8308), EEG-Inception (0.8829), and DeepConv (0.7735). Regarding the F1 score, our model achieved 88.54%.This comparison is described in the original 319-328.
Q5:
Applications and Future Research: Given the model's ability to adapt to new participants during inference, how can this approach be extended to other types of cognitive recognition tasks?
Thank you very much for your suggestions. By dynamically adjusting the filter length, the model can simultaneously capture features at different time scales. For example, in the CTP experimental paradigm, the model can be adapted to extract the relevant time segments for the P300 and N170 components. For other experimental paradigms, the model can be adjusted to extract different components based on the specific components elicited by the paradigm.
We added the above in lines 467-472.
Q6:
What additional research is needed to further enhance the model's generalization ability and computational efficiency?
Reply:
Thank you for your question. To extend our approach to other cognitive recognition tasks, the following steps can be taken:
We have added the following points in lines 472-479 to outline the additional research needed to further enhance the generalization capability and computational efficiency of the model, referencing the experiences of others in addressing related issues:
- 1. Collect and utilize larger and more diverse datasets encompassing a wide range of participants and cognitive tasks to enhance the model's generalization ability[37,38].
- 2. Improve the existing model by adapting the entire model structure to lightweight networks, such as Depthwise Separable Convolutions or Mixed Convolutions, to reduce computational complexity[39,40].
Reference
- Truong D, Sinha M, Venkataraju K U, et al. A streamable large-scale clinical EEG dataset for Deep Learning[C]//2022 44th Annual International Conference of the IEEE Engineering in Medicine & Biology Society (EMBC). IEEE, 2022: 1058-1061.
- Chen Y, Ren K, Song K, et al. EEGFormer: Towards Transferable and Interpretable Large-Scale EEG Foundation Model[J]. arXiv preprint arXiv:2401.10278, 2024.
- Sun W, Zhou X, Zhang X, et al. A lightweight neural network combining dilated convolution and depthwise separable convolution[C]//Cloud Computing, Smart Grid and Innovative Frontiers in Telecommunications: 9th EAI International Conference, CloudComp 2019, and 4th EAI International Conference, SmartGIFT 2019, Beijing, China, December 4-5, 2019, and December 21-22, 2019 9. Springer International Publishing, 2020: 210-220.
- Guo Z, Wang J, Chen S, et al. A Lightweight Stereo Matching Neural Network Based on Depthwise Separable Convolution[C]//2023 IEEE International Conference on Integrated Circuits, Technologies and Applications (ICTA). IEEE, 2023: 122-123.
Round 2
Reviewer 2 Report
Comments and Suggestions for Authors
I think the paper is ready to be published